**Comment**

# Optimising patient centred drug development to realise impact

Melanie Calvert, Paul Kamudoni, Christina Yap, Claire Snyder, Roger Wilson &
Olalekan Lee Aiyegbusi

Patient-reported outcomes can provide valuable evidence on the efficacy and tolerability of treatments throughout the drug development pipeline supporting regulatory decision-making, reimbursement and patient-centred care. Use of international guidelines and tools can optimise assessment and data quality and maximise impact.

Patient-reported outcomes (PROs) collected within drug development programmes can provide valuable evidence to inform shared decision making, labelling claims, clinical guidelines, and health policy.

Regulatory agencies, such as the US Food and Drug Administration (FDA) and European Medicines Agency (EMA) have released guidance to enhance the incorporation of the patient's voice in medical product development and regulatory decision making, whilst international societies and consortia have released methodological guidelines and resources to improve study design, reporting and implementation of findings. Here we summarise evidence on the value of PROs throughout the drug development pipeline and provide a summary of guidance documents to support the optimal PRO assessment in drug development (Fig. 1).

## Impact of PROs on drug development

Although traditionally collected in later phase clinical trials to assess the efficacy of treatments, PROs are increasingly being used in early phase clinical trials. PRO data has been used to confirm the tolerability of the recommended phase 2 dose (RP2D) and to guide dose escalation based on whether the maximum tolerated dose (MTD) was acceptable from a health-related quality-of-life (HRQOL) perspective[1].

PRO data can demonstrate the efficacy of drugs and support other clinical data. In later phase trials, as primary endpoints, PROs have been used to measure concepts such as palliation of symptoms, which are best assessed by patients[2,3]. A randomized trial in patients with symptomatic hormone-resistant prostate cancer compared prednisone with or without mitoxantrone using a PRO as primary outcome. Data collected using the McGill–Melzack Pain Questionnaire showed that mitoxantrone was associated with a sustained improvement in pain[3].

As secondary endpoints, PROs can support the findings of a clinical primary endpoint. For instance, in the phase 3, double-blind TOURMALINE-MM3 trial in post-transplant multiple myeloma patients, PRO data showed that, compared with placebo, ixazomib did not have an adverse impact on HRQOL, in addition to improving progression-free survival (PFS). Secondary PRO endpoints can also offer contrasting findings to the primary outcome[4]. For example, the Gynecologic Cancer Intergroup International Collaboration on Ovarian Neoplasms 7 (ICON7) trial, which evaluated adding bevacizumab to standard first-line chemotherapy, found that although the drug marginally improved the trial primary endpoint of PFS, it was associated with a clinically relevant negative impact on HRQOL compared with standard treatment at 54 weeks. In another trial, EORTC QLQ-C30 data was collected from women with advanced breast cancer on doxorubicin with or without vinorelbine. The PRO data showed improvements in patients' cancer pain and dyspnea suggesting palliative benefits[2].

The field is also increasingly focused on tolerability, a concept well-suited for PRO assessment. The COMPARZ (Pazopanib Versus Sunitinib in the Treatment of Locally Advanced and/or Metastatic Renal Cell Carcinoma) trial showed pazopanib to be noninferior to sunitinib for PFS and overall survival. However, the PRO data showed that pazopanib was better tolerated than sunitinib. The PRO data provided a more nuanced picture of treatment effects on fatigue and mouth-hand-foot symptoms than the summative toxicity data[4].

PRO data can inform regulatory drug approvals, labelling claims and reimbursement decisions. The tolerability data captured using PRO-Common Terminology Criteria for Adverse Events (PRO-CTCAE) for a noncontrolled trial of belantamab mafodotin-blmf were discussed during a FDA Oncologic Drugs Advisory Committee. It was noted in the FDA's accelerated approval letter that PROs should be collected in the confirmatory randomised controlled trial RCT[5]. The FDA also considered a PRO assessment of overall side effect impact from the FACT-GP5 data in a trial of adults and adolescents with advanced or metastatic RET-mutant medullary thyroid cancer. The FACT-GP5 data were supported by a lower incidence of treatment discontinuation due to adverse reactions for RETEVMO (selpercatinib) (4.7%) compared to cabozantinib or vandetanib (27%) in patients who received at least one dose of study treatment.

PRO data also informed FDA approval of ibrutinib for chronic graft vs host disease. The PRO data demonstrated efficacy, supporting the primary clinician-reported outcome, and was used in the FDA's benefit-risk assessment and included in the product labelling[5]. Other oncology drugs with PRO labelling granted by the FDA include abiraterone (for the treatment of prostate cancer), crizotinib (for the treatment of non-small cell lung cancer [NSCLC]), and ruxolitinib (for the treatment of myelofibrosis). A review by Gnanasakthy et al reported that between 2012 and 2016, there were 21 indications with PRO-related language in SmPCs (EMA PRO labelling), amounting to roughly one-third of all 64 indications reviewed by the EMA. PRO data from trials has been shown to directly inform national and international clinical guidelines and reimbursement decisions by Health Technology Assessment Agencies. For instance, the PRO data from the COMFORT-II trial provided evidence of symptom control that informed National Institute for Health and Care Excellence approval of ruxolitinib for myelofibrosis despite no statistically significant survival advantage.

**Fig. 1 | Shifting emphasis across the drug development continuum.** Tolerability is of primary focus in early phase trials, where it is critical for dose selection and understanding treatment side effects burden. Whilst treatment benefit becomes the dominant focus in later phase and real-world evidence (RWE) settings, tolerability remains a vital consideration throughout the therapeutic lifecycle.

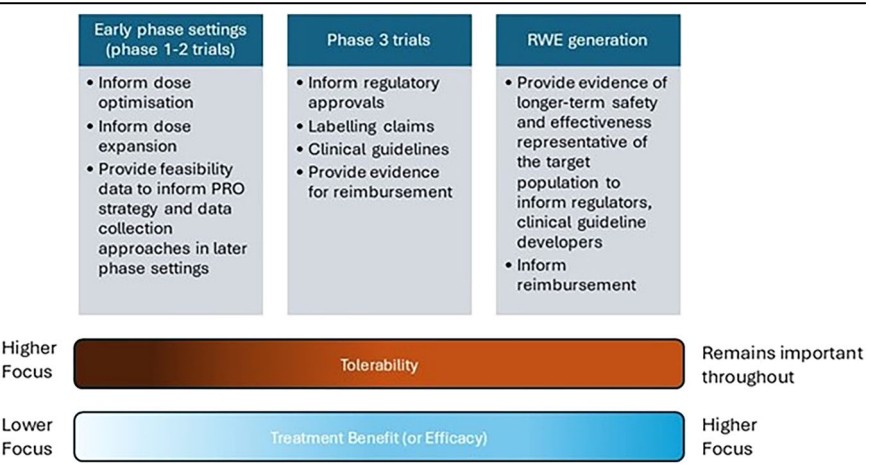

Whilst evidence of the impact of PROs for real world/post-marketing studies is presently limited, there is a steady increase in the utilisation of PROs in phase IV trials. Of 27,976 phase IV clinical trials posted on Clinicaltrials.gov between 1999 and July 2021, 21% and 4% used PROs and composite measures, respectively. Similarly, a review of EU post-authorization safety studies conducted between 2012 and 2015 showed that PROs were collected in 14% of Post-Authorization Safety Study (PASS) protocols and included assessments of symptoms, burden of disease and quality of life.

## Regulatory and international guidance to facilitate the inclusion of PROs in drug development

Maximising PRO result impact requires attention to unique considerations associated with measure selection, trial design, respondent burden, missing data, PRO score alerts, data capture and analysis, interpretation and presentation of results[4]. A range of international guidance and resources have been developed to support researchers and other key stakeholders throughout the drug development pipeline.

Guidance to support the optimal collection, analysis and reporting of PROs has been available for several decades, particularly in oncology where PROs offer valuable insights on the impact of cancer treatments to complement survival outcomes (overall, PFS).

A seminal moment was the publication of the 2009 FDA PRO guidance. Since then, there have been a series of PRO-related publications from the FDA, including the cancer core set, clinical outcome assessment (COA) compendium and the Patient-Focused Drug Development Guidance Series. The European Medicines Agency (EMA), have also published work in this area (EMA Appendix 2 to the guideline on the evaluation of anticancer medicinal products in man The use of patient-reported outcome (PRO) measures in oncology studies) and a more recent Patient experience data (PED) reflection paper. Alongside this, personnel from the FDA and EMA, the MHRA and Health Canada and NICE are also active contributors to many global initiatives to improve PRO data collection, analysis, reporting and implementation including SISAQOL, SPIRIT, and CONSORT. A limited number of these guidance documents are curated by the PROTEUS Consortium (Patient-Reported Outcomes Tools: Engaging Users and Stakeholders), which also offers educational videos, checklists and a handbook to support use. These international guidance and resources are summarised here so that trialists, researchers and other stakeholders can access relevant resources depending on their needs and the stage of the research (Table 1).

**Study design.** Most of the available guidance relates to study design. International guidelines have been developed on PRO-specific content to include in RCT protocols (SPIRIT-PRO), including resources for patient partners involved in co-design. Ethical considerations for the inclusion of PROs in clinical research have been provided, including a checklist to support submissions to ethics committees and more detailed recommendations to reduce respondent burden, manage PRO alerts, anonymise PRO data collected in clinical care for secondary use in research and address health equity considerations. Finally, tools are available to support inclusion of PROs in grant applications.

PRO measure selection is a key design consideration, and a recent review by members of the PROTEUS Consortium of International guidance on the selection of patient-reported outcome measures in clinical trials found general consistency across the recommendations. This work includes COSMIN guidance which has recently been updated. The ePROVIDE PROQOLID™ database is a comprehensive online database designed to assist academic researchers, physicians, students, pharmaceutical companies, health authorities, and international organizations search for and evaluate COAs, including PROs. Recommendations on the use of item libraries and tolerability measures, such as the PRO-CTCAE have also been provided. Given the increased use of electronic PROs (ePROs), several guidance documents address ePROs, including usability testing and system validation, measurement equivalence, data structure and standardization requirements to support drug development and training. Further considerations have been provided on the use of PROs in multi-site or multinational trials including the use of ePROs and translations and multi-site pragmatic trials using the electronic health record.

**Analysing data.** International standards for the analysis of quality-of-life and PRO endpoints in cancer RCTs have been developed by the SISAQOL Consortium and further developed by the SISAQOL-IMI Consortium. SISAQOL-IMI provides guidance on implementing PROs in RCTs and single arm trials, including PRO score interpretation thresholds, PRO data visualisation and terminology. Although developed for oncology, these resources may also be useful for other clinical areas.

**Table 1 | Guidance to support the use of PROs in drug development programmes**

| Study Design | Writing Grants |
| --- | --- |
| | Recommendations for including or reviewing patient reported outcome endpoints in grant applications |
| | Developing Protocols/Study Design Considerations |
| | Guidelines for Inclusion of Patient-Reported Outcomes in Clinical Trial Protocols: The SPIRIT-PRO Extension |
| | SPIRIT-PRO Extension explanation and elaboration: guidelines for inclusion of patient-reported outcomes in protocols of clinical trials |
| | 'Give Us The Tools!': development of knowledge transfer tools to support the involvement of patient partners in the development of clinical trial protocols with patient-reported outcomes (PROs), in accordance with SPIRIT-PRO Extension \| BMJ Open |
| | Recommendations for incorporating patient-reported outcomes into clinical comparative effectiveness research in adult oncology |
| | Checklist for Clinical Research Professionals |
| | Systematic collection of patient reported outcome research data: A checklist for clinical research professionals |
| | Multi-site/International Trials |
| | Design and analytic considerations for using patient-reported health data in pragmatic clinical trials: report from an NIH Collaboratory roundtable |
| | Considerations when introducing electronic patient-reported outcome data capture in multicentre oncology randomised controlled trials |
| | Multinational trials-recommendations on the translations required, approaches to using the same language in different countries, and the approaches to support pooling the data: the ISPOR Patient-Reported Outcomes Translation and Linguistic Validation Good Research Practices Task Force report |
| | ePROs |
| | Key methodological considerations for usability testing of electronic patient-reported outcome (ePRO) systems |
| | Validation of electronic systems to collect patient-reported outcome (PRO) data-recommendations for clinical trial teams: report of the ISPOR ePRO systems validation good research practices task force |
| | Updated Recommendations on Evidence Needed to Support Measurement Comparability Among Modes of Data Collection for Patient-Reported Outcome Measures: A Good Practices Report of an ISPOR Task Force |
| | Best Practice Recommendations for Electronic Patient-Reported Outcome Dataset Structure and Standardization to Support Drug Development |
| | Training on the Use of Technology to Collect Patient-Reported Outcome Data Electronically in Clinical Trials: Best Practice Recommendations from the ePRO Consortium |
| | Measurement Equivalence of Patient-Reported Outcome Measures Migrated to Electronic Formats: A Review of Evidence and Recommendations for Clinical Trials and Bring Your Own Device |
| | Considerations for Requiring Subjects to Provide a Response to Electronic Patient-Reported Outcome Instruments |
| | Health Equity |
| | Health Equity Considerations for Developing and Reporting Patient-reported Outcomes in Clinical Trials: A Report from the OMERACT Equity Special Interest Group |
| | Patient reported outcome assessment must be inclusive and equitable |
| | Ethical considerations |
| | Ethical Considerations for the Inclusion of Patient-Reported Outcomes in Clinical Research: The PRO Ethics Guidelines |
| | Patient-reported outcome alerts: ethical and logistical considerations in clinical trials |
| | Recommendations to address respondent burden associated with patient-reported outcome assessment |
| | Montreal Accord on Patient-Reported Outcomes (PROs) use series - Paper 9: anonymization and ethics considerations for capturing and sharing patient reported outcomes |
| | Selecting measures |
| | International guidance on the selection of patient-reported outcome measures in clinical trials: a review |
| | ePROVIDE |

**Study reporting.** CONSORT-PRO aims to promote transparent reporting of trials in which PROs are primary or important secondary outcomes. Guidance on the graphical presentation of findings has also been developed. Specific reporting guidelines are also available for the European Society for Medical Oncology-Magnitude of Clinical Benefit Scale credit and HealthMeasures' PROs (ASCQ-Me, Neuro-QoL, NIH Toolbox, PROMIS).

**Applying findings.** Once reported, PRO data can inform shared decision making, clinical guideline development, labelling claims and health policy.

**Table 1 (continued) | Guidance to support the use of PROs in drug development programmes**

| | | |
|---|---|---|
| Study Design Continued | Systematic Reviews of PRO measures | |
| | COSMIN guideline for systematic reviews of patient-reported outcome measures version 2.0 | |
| | COSMIN reporting guideline for studies on measurement properties of patient reported outcome measures | |
| | Item Libraries | |
| | Recommendations on the use of item libraries for patient-reported outcome measurement in oncology trials: findings from an international, multidisciplinary working group | |
| | Tolerability | |
| | Methodological standards for using the patient-reported outcomes version of the common terminology criteria for adverse events (PRO-CTCAE) in cancer clinical trials | |
| | Patient-Reported Outcome Measures in Safety Event Reporting: PROSPER Consortium guidance | |
| | Broadening The Definition Of Tolerability In Cancer Clinical Trials To Better Measure The Patient Experience | |
| Analysing data | International guidance | |
| | Setting International Standards in Analyzing Patient-Reported Outcomes and Quality of Life Endpoints in Cancer Clinical Trials-Innovative Medicines Initiative (SISAQOL-IMI): stakeholder views, objectives, and procedures | |
| | SISAQOL-IMI consensus-based guidelines to design, analyse, interpret, and present patient-reported outcomes in cancer clinical trials | |
| Reporting Data | Trial Publications | |
| | Reporting of patient-reported outcomes in randomized trials: the CONSORT PRO extension | |
| | Patient-reported outcomes in randomized clinical trials: development of ISOQOL reporting standards | |
| | Specific Reporting Standards | |
| | Methodological and reporting standards for quality-of-life data eligible for European Society for Medical Oncology-Magnitude of Clinical Benefit Scale (ESMO-MCBS) credit | |
| | A reporting checklist for HealthMeasures' patient-reported outcomes: ASCQ-Me, Neuro-QoL, NIH Toolbox, and PROMIS | |
| | Graphical presentation | |
| | Making a picture worth a thousand numbers: recommendations for graphically displaying patient-reported outcomes data | |
| Applying Findings | Clinical use | |
| | Clinician's checklist for reading and using an article about patient-reported outcomes | |
| | Drug Advisory Committee Meetings | |
| | A Review of Patient-Reported Outcome Considerations in Oncologic Drugs Advisory Committee Meetings (2016-2021) | |
| Context-specific Considerations | Early phase trials | |
| | Assessment of Patient-Reported Outcomes in Industry-Sponsored Phase I Oncology Studies: Considerations for Translating Theory Into Practice | |
| | Advancing patient-centric care: integrating patient reported outcomes for tolerability assessment in early phase clinical trials – insights from an expert virtual roundtable | |
| | International Consensus-Driven Recommendations for Patient-Reported Outcome Research Objectives in Early Phase Dose-Finding Oncology Trials: OPTIMISE-ROR | |
| | A practical toolkit with recommendations for analysing and visualising patient-reported outcomes in early phase dose-finding oncology trials (OPTIMISE-AR) | |
| | Real-World Evidence Generation | |
| | Designing and Implementing Real World Patient Reported Outcomes (RW- PROs) - Emerging Recommendations: A Good Practices Report of an ISPOR Task Force | |
| | Children and adolescents | |
| | Applying a developmental approach to quality of life assessment in children and adolescents with psychological disorders: challenges and guidelines | |

A checklist has been developed to support the use of PROs by clinicians, and a review of PRO considerations in Oncologic Drugs Advisory Committee Meetings provides insights on the use of PRO data in regulatory decision making.

**Context-specific guidance.** Further context-specific guidance has been developed including use of PROs in early phase trials, for real-world evidence generation and in children and adolescents.

**Table 1 (continued) | Guidance to support the use of PROs in drug development programmes**

| Regulatory Guidance and Associated Commentaries | FDA |
|---|---|
| | FDA Draft Guidance for Industry on Core Patient-Reported Outcomes in Cancer Clinical Trials |
| | International Society for Quality of Life Research commentary on the US Food and Drug Administration draft guidance for industry on core patient-reported outcomes in cancer clinical trials |
| | FDA COA Compendium |
| | Patient-Reported Outcome Measures in the Food and Drug Administration Pilot Compendium: Meeting Today's Standards for Patient Engagement in Development? |
| | FDA 2009 Guidance for Industry Patient-Reported Outcome Measures: Use in Medical Product Development to Support Labeling Claims |
| | Patient-Focused Drug Development Guidance Series |
| | EMA |
| | EMA Appendix 2 to the guideline on the evaluation of anticancer medicinal products in man The use of patient-reported outcome (PRO) measures in oncology studies |

**Implementation of results and maximising impact.** Several barriers associated with the implementation of PROs in clinical trials may hamper the impact of the data. Adherence to the guidelines recommended in this Comment and multi-stakeholder collaboration are essential to maximise the use of PRO trial data, facilitate impact and minimise research waste.

While it is difficult to isolate the direct impact of international guidelines on the number of trials using PROs and the quality of trial design and reporting, there is evidence to suggest that citing CONSORT-PRO is associated with higher quality reporting[6]. There is also an increasing trend in use of clinical outcomes assessments, including PROs in FDA labels and health technology assessment submissions, in total and across most therapeutic areas, indicating a shift toward more patient centricity[7].

## Data availability

The full list of screened sources and the data extraction framework used for this scoping review are available from the corresponding author upon reasonable request.

Melanie Calvert [1,2,3,4,5] ✉, Paul Kamudoni[6], Christina Yap [7], Claire Snyder[8], Roger Wilson [9] & Olalekan Lee Aiyegbusi [1,2,3,4,5]

[1]Centre for Patient Reported Outcomes Research, Department of Applied Health Sciences, College of Medicine and Health, University of Birmingham, Birmingham, UK. [2]Birmingham Health Partners Centre for Regulatory Science and Innovation, University of Birmingham, Birmingham, UK. [3]National Institute for Health and Care Research (NIHR) Birmingham Biomedical Research Centre, University of Birmingham, Birmingham, UK. [4]National Institute for Health and Care Research (NIHR) Applied Research Collaboration (ARC) West Midlands, University of Birmingham, Birmingham, UK. [5]NIHR Blood and Transplant Research Unit (BTRU) in Precision Transplant and Cellular Therapeutics, University of Birmingham, Birmingham, UK. [6]Merck KGaA, Darmstadt, Germany. [7]Clinical Trials and Statistics Unit, The Institute of Cancer Research, London, UK. [8]Johns Hopkins Schools of Medicine and Public Health, Baltimore, MD, USA. [9]Patient Member, Cancer Research Advocates Forum UK, London, UK. ✉e-mail: m.calvert@bham.ac.uk

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

## Acknowledgements

M.J.C. receives funding from the NIHR Birmingham Biomedical Research Centre, NIHR Surgical Reconstruction and Microbiology Research Centre, NIHR Blood and Transplant Research Unit (BTRU) in Precision Transplant and Cellular Therapeutics, and NIHR ARC West Midlands at the University of Birmingham and University Hospitals Birmingham NHS Foundation Trust, LifeArc, Health Data Research UK, Innovate UK (part of UK Research and Innovation), Macmillan Cancer Support, European Regional Development Fund – Demand Hub, SPINE UK, UKRI, UCB Pharma, GSK, Anthony Nolan, and Gilead Sciences.C.S. currently receives unrestricted education grants through her institution from Pfizer as PI of the PROTEUS Consortium. The funder had no role in the conceptualization, design, data collection, analysis, decision to publish, or preparation of the manuscript.O.L.A. receives funding from the NIHR Birmingham Biomedical Research Centre (BRC), NIHR Applied Research Collaboration (ARC), West Midlands, NIHR Blood and Transplant Research Unit (BTRU) in Precision Transplant and Cellular Therapeutics at the University of Birmingham and University Hospitals Birmingham NHS Foundation, LifeArc, Innovate UK (part of UK Research and Innovation), The Health Foundation, Gilead Sciences Ltd, Merck, Anthony Nolan, GSK, and Sarcoma UK.This work is funded by Merck Healthcare.

## Author contributions

M.J.C., P.K., and O.L.A. conceptualised the review. M.J.C., C.Y., and O.L.A. wrote and edited the manuscript. M.J.C., P.K., O.L.A., C.Y., C.S., and R.W. reviewed the manuscript. M.J.C., P.K., O.L.A., C.Y., C.S., and R.W. approved the final version.

## Competing interests

MJC received personal fees from Astellas, Boehringer Ingelheim, Aparito Ltd, CIS Oncology, Gilead, Halfloop, Takeda, Merck, Daiichi Sankyo, Glaukos, GSK, Pfizer, Vertex, Shionogi, and the Patient-Centered Outcomes Research Institute (PCORI) outside the submitted work. In addition, a family member owns shares in GSK. C.Y. report personal fees from Faron Pharmaceuticals, Bayer, Trogenix and Merck, outside the submitted work. C.S. currently receives personal consulting fees from Movember and Pfizer and previously received personal consulting fees from Shionogi. Dr. Snyder has received travel costs for meeting presentations from Shionogi and Executive Insight Healthcare

Consultants. She also receives research funding from Johnson & Johnson through her institution. O.L.A. declares personal fees from Gilead Sciences, Merck, Boehringer Ingelheim, Innovate UK, and GSK. All other authors declare no competing interests.
