## [Transparent Peer Review file · Communications Medicine]

Optimising patient centred drug development to realise impact

Corresponding Author: Professor Melanie Calvert

Version 0:

Reviewer comments:

Reviewer #1

(Remarks to the Author)

General comments:

Thank you for the opportunity to review this paper, aiming to summarise evidence on the value of PROs throughout the drug development pipeline and provide a summary of guidance documents to support the optimal PRO assessment in drug development. It provides a very comprehensive and useful summary that should be of great value to the research community.

Impact of PROs on drug development

Line 49: Remove "a" before "methodological"

The second section (from line 61-79) has several very interesting examples of trials where PROs have had substantial value. The section is a bit hard to read, though. For example, the example described in line 72-74 seems more appropriate earlier in this section. Perhaps the authors can structure the section a bit differently, for example so that trials with PROs as primary endpoints are mentioned before those where PROs were secondary endpoints?

EMA should be spelled out in the introduction (it's spelled out in the Regulatory and international guidance to facilitate the inclusion of PROs in drug development section).

Regulatory and international guidance to facilitate the inclusion of PROs in drug development

The authors list an impressive list of guidelines and resources that can aid the reader. If possible, some idea of the value of these publications could be added to the end of the article. For example, has it increased the number of trials where PROs are the primary outcome, or has it improved other metrics of interest?

Reviewer #2

(Remarks to the Author)

This paper was strange to read as it is a narrative summary of different guidance documents on the planning, conduct, or interpretation of PROs. I don't really see what the point of this article is. It is neither a systematic review nor poses a question and then attempts to answer it.

Version 1:

Reviewer comments:

Reviewer #1

(Remarks to the Author)

Thank you for the opportunity to review the revised manuscript. I believe the authors have enhanced the manuscript and present a useful and comprehensive overview of the PRO field. I have no further comments to add.

Response to Editor and Reviewers

Manuscript Title: Optimising patient-centred drug development to realise impact

Journal: Communications Medicine

We thank the editor and reviewers for their time and constructive feedback. We have revised the manuscript to improve clarity, flow, and the logical presentation of trial examples.

Below are our point-by-point responses.

Reviewer #2

General Comment: Thank you for the opportunity to review this paper... It provides a very comprehensive and useful summary that should be of great value to the research community.

Response: We thank the reviewer for their positive assessment and for recognising the value of this summary for the research community.

Comment 1: Line 49: Remove “a” before “methodological”

Response: We have removed the word "a" as suggested.

Comment 2: The second section (from line 61-79) has several very interesting examples... Perhaps the authors can structure the section a bit differently, for example so that trials with PROs as primary endpoints are mentioned before those where PROs were secondary endpoints?

Response: We agree that this improves the narrative flow. We have restructured this section to first discuss trials where PROs served as primary endpoints, followed by those where they provided critical secondary evidence. This better illustrates the hierarchy of clinical evidence and the varying roles PROs play in regulatory success.

Comment 3: EMA should be spelled out in the introduction.

Response: We have spelled out the European Medicines Agency (EMA) in the introduction and have also spelt out FDA.

Comment 4: If possible, some idea of the value of these publications [guidelines] could be added to the end of the article. For example, has it increased the number of trials where PROs are the primary outcome?

Response: This is an excellent point; we have added a concluding paragraph. It is difficult to directly attribute changes in practice, for example use of PROs as a primary outcome to guidelines. However, we have cited recent work demonstrating an increased trend in use of clinical outcomes assessments, including PROs in FDA labels and health technology assessment submissions, in total and across most therapeutic areas, indicating a shift toward more patient centricity

Reviewer #3:

Comment 1: This paper was strange to read as it is a narrative summary of different guidance documents... I don't really see what the point of this article is. It is neither a systematic review nor poses a question and then attempts to answer it.

Response: We appreciate the reviewer's perspective; however, we would like to clarify the intended purpose of this piece as a "Comment" manuscript, formatted for the journal.

The primary objective is not to conduct a systematic review, but to provide a curated, high-level synthesis of a complex and rapidly evolving regulatory landscape. For many researchers and clinical trialists, the sheer volume of international guidance can be a barrier to implementation. This article serves as a strategic roadmap, identifying the most critical frameworks to ensure patient-centred data is "regulatory-ready."

By distilling these guidelines and providing real-world examples of their impact, we aim to bridge the gap between methodological theory and clinical trial practice. We believe this "translation" of guidance into actionable insights is of high value to the readership of Communications Medicine.

Thank you for considering our work for publication.

Kind Regards
Professor Melanie Calvert